# StructZip: Compressing Large-Scale Structured Prompts to One Token via Learning Natural Language Descriptions

## Abstract

Tool use has become a central capability in large language model (LLM)-based agents, enabling them to interact with external environments through structured APIs. However, effective tool use typically requires including a large number of tool descriptions, often with complex schemas, in the context for each inference. This static and structured portion of the prompt grows linearly with the number of tools and poses a significant challenge to inference efficiency. Although prior work has explored prompt compression for long contexts, most approaches focus on unstructured text and are not optimized for the compression of structured prompts. To bridge this gap, we introduce **StructZip**, a novel framework that transforms large structured prompts into parametric memory, which can be elicited by a single token. Our approach first "unzips" the structured prompt into a set of semantically equivalent question-answer (QA) pairs. By fine-tuning the LLM on these QA pairs, StructZip encodes the information into the model's parameters, making it accessible through a designated special token at inference time. We evaluate our method on three representative tasks: table-based question answering, tool-use, and closed-set text classification. Experimental results demonstrate that StructZip can compress prompts of millions of tokens into a single one while maintaining performance nearly on par with using the full, uncompressed prompts, offering a practical solution for efficient structured data handling in LLMs.

## 1 Introduction

The advent of large language models (LLMs) has significantly advanced the capabilities of AI agents, enabling them to tackle increasingly complex tasks by reasoning, planning, and interacting with external environments. However, a critical challenge hindering their broader application is the processing of extensive structured data, such as detailed tool descriptions, large tables, or complex classification taxonomies. The naive approach of concatenating this data directly into the prompt is often infeasible. For instance, a comprehensive set of API documentation can easily exceed thousands of tokens, leading to prohibitive inference costs and latency, and frequently surpassing the context length limitations of even the most advanced models.

To address the challenges posed by long prompts, prior research has explored various strategies. Some approaches focus on architectural modifications to better handle extended contexts Kitaev et al. (2020); Zhou et al. (2021). Others investigate prompt compression, where methods like LLM-Lingua Jiang et al. (2023a) and 500xCompressor Li et al. (2024) have demonstrated success in compressing unstructured textual prompts. However, these techniques are fundamentally ill-suited for structured data. The high information density and rigid syntax of formats like JSON mean they possess minimal redundancy. Unlike natural language, altering or omitting even a single token can corrupt the data's integrity, leading to catastrophic parsing failures or silent reasoning errors during inference. This leaves a critical research gap: an effective compression method for structured prompts that preserves their semantic and structural integrity.

To bridge this gap, we introduce **StructZip**, a novel method that compresses large structured prompts into a single token. Inspired by prior work on knowledge representation Dong et al. (2017);

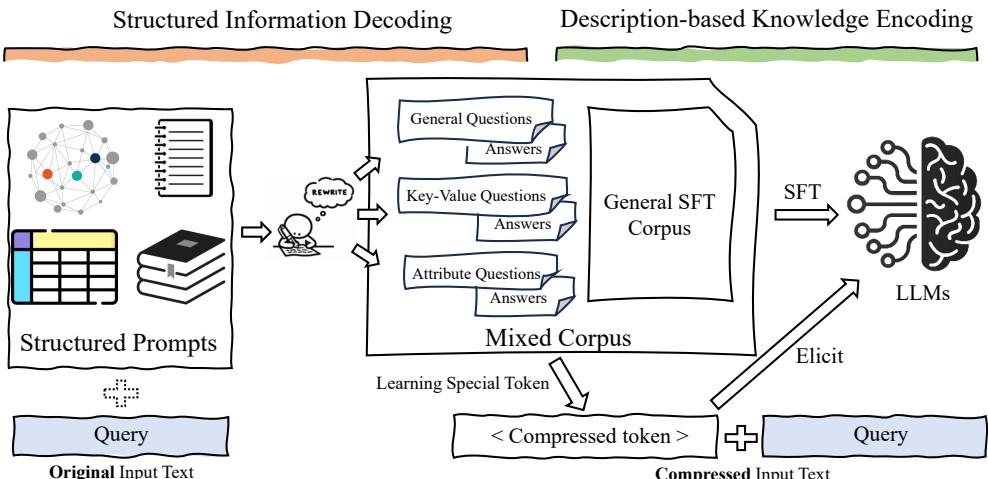

Figure 1: Overview of the StructZip framework. A structured prompt is first "unzipped" into semantically equivalent Q-A pairs (Structured Information Decoding). These pairs are then mixed with generic SFT data to fine-tune the LLM (Description-based Knowledge Encoding). At inference time, the embedded knowledge can be elicited by a single compressed token.

Zhu et al. (2019); Lewis et al. (2019); Min et al. (2024), our core idea is to create a set of natural language question-answer (QA) pairs that are semantically equivalent to the original structured data. Instead of embedding the bulky data directly, we fine-tune the LLM on these QA pairs. This training process encodes the explicit, structured information into the model's parametric memory Du et al. (2025); Wu et al. (2025b). A designated compressed token, trained alongside this process, serves as a compact key. During inference, this single token is used to elicit the embedded knowledge, allowing the model to reason over the complete information as if it were fully present in the context.

To validate our approach, we conduct extensive experiments on three representative tasks: table-based question answering, tool-use, and closed-set text classification. Our results demonstrate that StructZip achieves extreme compression ratios, i.e., reducing prompts of millions of tokens to a single one, while maintaining performance nearly on par with using the full, uncompressed prompts. It significantly outperforms existing compression baselines in structured data scenarios, highlighting its effectiveness and practical value for developing more efficient and scalable LLM-based agents.

## 2 PROBLEM DEFINITION

We formally define the problem of **Large Structured Prompt Compression (LSPC)**. This problem addresses a class of prompts characterized by three key properties. First, they are **extremely long**, often exceeding the context window limits of large language models (LLMs). Even when they fit, their length leads to prohibitive inference costs and high latency. Second, the information is **highly dense** with minimal redundancy, meaning conventional lossy compression techniques would cause substantial information loss. Third, they are **highly formatted** with a rigid structure (e.g., JSON schemas, tables), where even minor alterations can corrupt their integrity and lead to reasoning failures. The objective of LSPC is to enable LLMs to process these prompts while preserving their informational content, without incurring token costs at inference time.

The LSPC problem formulation is highly general, encompassing various tasks that rely on structured data. In this work, we demonstrate the broad applicability of our approach by focusing on three representative and challenging tasks: **Table-based Question Answering**, which involves compressing large, detailed tables for querying; **Tool-Use**, for compressing extensive API documentation for agentic systems; and **Closed-set Text Classification**, which requires compressing a large set of class labels and their detailed descriptions. These tasks represent common yet difficult scenarios where structured prompts are essential, and we believe the LSPC framework can be extended to an even wider range of future applications involving structured data.

# 3 METHOD

To address the LSPC problem, we introduce **StructZip**, a novel framework that transforms large structured prompts into parametric memory Du et al. (2025); Wu et al. (2025b) within an LLM which can be elicited by a single token. This is achieved through a two-stage decoding-and-encoding process. As illustrated in Figure 1, the structured prompt is first "unzipped" into a set of natural language question-answer (QA) pairs, which could maintain all the information of the original structured prompt. The core intuition driving this design, is that natural language descriptions can serve as a faithful and comprehensive proxy for complex structured data. These QA pairs are then used to fine-tune the model, encoding the information into its parameters. A designated compressed token, trained alongside this process, acts as the key to access this newly formed memory in the inference stage.

## 3.1 STRUCTURED INFORMATION DECODING

The first stage of StructZip, Structured Information Decoding, is responsible for "unzipping" the dense, structured prompt into a comprehensive set of natural language question-answer (QA) pairs. This process is designed to be fully reversible, ensuring that the complete semantic and structural information of the original data is preserved. To achieve this, we generate a diverse range of QA pairs that probe the structured data from multiple perspectives. For example, when compressing a text classification system, we generate the following QA pairs:

- **General Questions**, which query for the entirety of the structured data to provide a holistic view.

```
1  {
2    "prompt": "What is <|data|>?",
3    "answer": "{{output the entire system}}"
4  }
5  {
6    "prompt": "Based on <|data|>, output all category names.",
7    "answer": "{{output the entire system}}"
8  }
```

- **Key-Value Questions**, which target specific content points.

```
1  {
2    "prompt": "is the label A in the <|data|>?",
3    "answer": "Yes"
4  }
```

- **Attribute Questions**, which inquire about metadata and properties of the structure, such as quantity and qualitative descriptions.

```
1  {
2      "prompt": "How many categories are there in total in the <|
           data|>?",
3      "answer": "263"
4  }
5
6  {
7      "prompt": "How many subcategories are there in category D
           based the <|data|>?",
8      "answer": "3"
9  }
```

This decoding strategy is universally applicable across different data formats: it translates table rows into factual statements, API function signatures into capability descriptions, and classification taxonomies into hierarchical queries. Within the questions, a placeholder token (e.g., `<|data|>`) is used to conceptually refer to the structured prompt being described.

### 3.2 Description-based Knowledge Encoding

The second stage, Description-based Knowledge Encoding, embeds the knowledge from the decoded QA pairs into the LLM's parameters. To achieve this while preserving the model's general capabilities, we create a new training corpus by mixing our generated QA pairs with an existing, general-purpose Supervised Fine-Tuning (SFT) dataset such as the SFT dataset used in LlaMa-3 Grattafiori et al. (2024). The model is then fine-tuned on this composite dataset using a standard SFT procedure. This process trains the model to associate the compressed token (using special token like `<|data|>`) with the complete information of the original structured prompt, effectively compiling the explicit, lengthy data into a compact, implicit parametric memory.

**Inference** Consequently, the entire structured prompt can be substituted with this single compressed token during inference. This single compressed token allows the model to elicit the stored knowledge with zero additional token overhead while response to the queries.

## 4 Experiments

This paper discusses the compression of structured or semi-structured prompts. To the best of my knowledge, this issue has not been explored in previous work, and there are no directly comparable benchmarks. We selected three typical scenarios: table-based question answering, tool invocation, and text classification. We adapted the relevant benchmarks by modifying the prompts, but this adaptation does not affect the fairness of the evaluation.

### 4.1 Datasets

#### 4.1.1 Text Classification

**TNEWS** [1] is a traditional text classification task. The dataset consists of Chinese news articles published by TouTiao before May 2018, with a total of 73,360 titles. Each title is labeled with one of 15 news categories (such as finance, technology, sports, etc.), and the task is to predict which category the title belongs to. The data is in Chinese language and is stored in a JSON file format containing 73,360 entries.

**English Dolly 2.0** Conover et al. (2023) databricks-dolly-15k is a corpus of more than 15,000 records generated by thousands of Databricks employees to enable large language models to exhibit the magical interactivity of ChatGPT. Databricks employees were invited to create prompt/response pairs in each of eight different instruction categories, including the seven outlined in the InstructGPT paper Ouyang et al. (2022), as well as an open-ended free-form category. The contributors were instructed to avoid using information from any source on the web except for Wikipedia (for particular subsets of instruction categories), and were explicitly instructed to avoid using generative AI in formulating instructions or responses. Examples of each behavior were provided to motivate the types of questions and instructions appropriate to each category.

**Chinese FireflyConover et al. (2023)** We have collected 23 common Chinese datasets. For each task, several instruction templates were manually written to ensure the high quality and richness of the data, totaling 1.15 million entries. To make it comparable to the English dataset, we randomly sampled 15k data points from it.

**Setting** We performed the same preprocessing on each dataset before training. First, we collect labels for the current dataset. After collecting the labels, each label is annotated to form a classification system. Ultimately, this classification system will be concatenated with each question as a

---

[1]https://github.com/fatecbf/toutiao-text-classfication-dataset/

| Method | | Acc | Context | Input | Output | Ratio | Ini-lat.(ms) | Lat.(ms) |
|---|---|---|---|---|---|---|---|---|
| **Chinese Text Classification** | | | | | | | | |
| TNEWS | | | | | | | | |
| GPT4o | | 0.722 | 514 | 606 | / | 1x | | |
| LongLLMLingua | | 0.606 | 321 | 414 | 38.42 | 1.6x | | |
| AutoCompressors | | 0.554 | 102 | 195 | 67.22 | 5x | / | |
| Gist | | 0.656 | 128 | 221 | 55 | 4x | | |
| 500xCompressor | | 0.702 | 128 | 221 | 46 | 4x | | |
| StructZip | w/o | 0.905 | 514 | 606 | 2 | 1x | 70.13 | 118.74 |
| (Qwen2.5-7B) | w/ | 0.903 | 107 | 275 | 2 | 4.8x | 60.22 | 109.04 |
| Firefly | | | | | | | | |
| GPT4o | | 0.850 | 5062 | 5219.90 | / | 1x | | |
| LongLLMLingua | | 0.652 | 3164 | 3321 | 180.4 | 1.6x | | |
| AutoCompressors | | 0.664 | 1012 | 1170 | 220.3 | 5x | / | |
| Gist | | 0.723 | 1265 | 1423 | 145.1 | 4x | | |
| 500xCompressor | | 0.824 | 1265 | 1423 | 148.9 | 4x | | |
| StructZip | w/o | 0.890 | 5062 | 5219 | 124.2 | 1x | 429.8 | 6252.61 |
| (Qwen2.5-7B) | w/ | 0.887 | 335 | 570 | 121.6 | 15.1x | 62.38 | 5547.76 |
| **English Text Classification** | | | | | | | | |
| Dolly 2.0 | | | | | | | | |
| GPT4o | | 0.714 | 682 | 824 | / | 1x | | |
| LongLLMLingua | | 0.432 | 426 | 569 | 77.1 | 1.6x | | |
| AutoCompressors | | 0.501 | 136 | 279 | 89.3 | 5x | / | |
| Gist | | 0.608 | 170 | 312 | 67.2 | 4x | | |
| 500xCompressor | | 0.668 | 170 | 312 | 32.4 | 4x | | |
| StructZip | w/o | 0.753 | 682 | 824 | 3.6 | 1x | 76.78 | 230.55 |
| (Llama3.1-8B) | w | 0.754 | 222 | 367 | 3.6 | 3.1x | 76.36 | 230.00 |
| **Table Question Answering** | | | | | | | | |
| TabelBench | | | | | | | | |
| GPT4o | | 0.743 | 885♣ | 1168 | / | 1x | | |
| LongLLMLingua | | 0.332 | 553 | 836 | 66.1 | 1.6x | | |
| AutoCompressors | | 0.271 | 177 | 460 | 78,3 | 5x | / | |
| Gist | | 0.504 | 221 | 504 | 63.2 | 4x | | |
| 500xCompressor | | 0.607 | 221 | 504 | 60.6 | 4x | | |
| StructZip | w/o | 0.655 | 885♣ | 1168 | 52.3 | 1x | 147.59 | 2511.17 |
| (Qwen2.5-7B) | w | 0.649 | 68 | 65 | 51.8 | 13.1x | 61.26 | 2335.44 |
| **Tool Invocation** | | | | | | | | |
| XLAM | | | | | | | | |
| GPT4o | | 0.983 | | 1346 | / | 1x | | |
| LongLLMLingua | | 0.412 | | 2155 | 349.4 | 1.6x | | |
| AutoCompressors | | 0.322 | $3M$♠ | 6734 | 325.3 | 5x | / | |
| Gist | | 0.378 | | 5387 | 450.3 | 4x | | |
| 500xCompressor | | 0.456 | | 5387 | 308.2 | 4x | | |
| StructZip | w/o | 0.982 | | 1346 | 217.6 | 1x | 94.83 | 13239.97 |
| (Llama3.1-8B) | w | 0.945 | 225 | 329 | 214.7 | $13.3kx$ | 81.66 | 13189.27 |

Table 1: Here are our experimental results across three different tasks. "w/o" indicates the absence of compression methods, meaning the context is directly concatenated with the instruction. "w/" refers to the use of the compression methods discussed in this paper. For GPT-4o, due to the inability to train, all results pertain to prompts without any compression method applied. The length metrics provided represent average lengths. It is particularly notable that in tool invocation scenarios, there are over 30,000 tool descriptions, which far exceed the model's length capacity, making direct concatenation impossible. Therefore, both GPT-4o and "w/o" conditions involve direct concatenation of the 20 actual tool descriptions, representing the ideal situation.

prompt. For example, for the "text summarization" label, the annotation would be: "Summarize the text into a short paragraph that captures the main points of the entire text." Detailed annotations are listed in the appendix, and the full system will be open-sourced.

### 4.1.2 TABLE QUESTION ANSWERING

**TableBench** Wu et al. (2025a) TableBench is a comprehensive benchmark designed to evaluate large language models' (LLMs) capabilities in table question answering (TableQA) across 18 fields within four major categories: fact-checking, numerical reasoning, data analysis, and visualization. It comprises 886 test samples that challenge LLMs with complex reasoning tasks involving tabular data. Additionally, TableBench introduces TableLLM, a model trained on the TableInstruct dataset, which achieves performance comparable to GPT-3.5. Extensive experiments indicate that

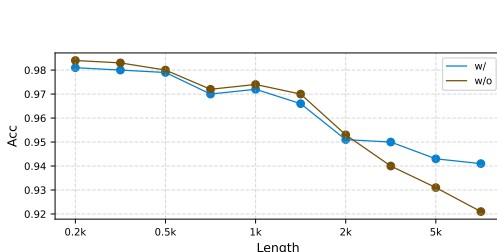 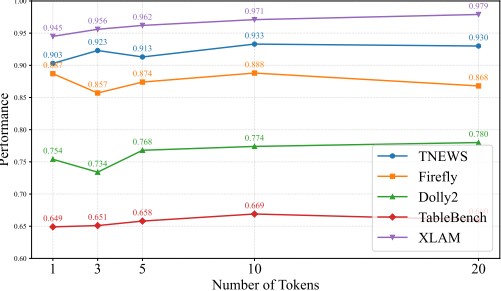

(a) The effect of compression on prompt length and complexity on XLAM dataset.

(b) The results of using different numbers of compressed tokens on different tasks.

Figure 2: Left: prompt length and complexity vs. compression; Right: number of compressed tokens vs. performance.

both open-source and proprietary LLMs have significant room for improvement in handling real-world TableQA scenarios, with even advanced models like GPT-4 achieving only modest scores compared to human performance.

**Setting** During the training phase, we constructed table description corpora for all the tables used in the test set using the method outlined in Section 2.1. Due to limited training resources, we randomly selected 300,000 samples from TableInstruct and mixed them with the table description corpora as the fine-tuning dataset. During the testing phase, we followed the approach described in the paper for evaluation, but when concatenating tables, we referred to the descriptions used in the classification tasks.

### 4.1.3 TOOLS INOVCATION

**xlam-function-calling-60k** Liu et al. (2024b) dataset is a collection designed to facilitate research in code generation and function calling tasks. It contains 60,000 examples of function call patterns, where each example pairs a natural language description with the corresponding function call in programming languages. The dataset aims to support the training and evaluation of models that translate natural language into executable code, particularly focusing on how well models can understand and generate function calls based on given instructions. The diverse set of examples enables the development of more robust code generation tools and benchmarks for evaluating language models' performance in programming tasks.

### 4.2 BASELINES

This paper addresses the issue of structured compression, a topic that has not been specifically discussed in previous work. We selected LongLLMlinguaJiang et al. (2023a), AutoCompressorsChevalier et al. (2023), GistMu et al. (2023), and 500xCompressorLi et al. (2024), which represent both hardware and software compression methods related to prompt compression. Additionally, we used an uncompressed setting as a control for comparison experiments.

### 4.3 MAIN RESULTS

Table1 presents evaluation results across five datasets spanning three representative tasks. We report standard performance metrics including accuracy, context length, total input and output lengths, compression ratio, first-token latency, and total latency. Overall, our method consistently outperforms existing compression techniques across all tasks. Notably, the performance of our compressed inputs is on par with, and in some classification cases even surpasses, the results of uncompressed inference with GPT-4o.

For text classification, both in Chinese and English settings, we observe that models fine-tuned on compressed inputs yield better accuracy than zero-shot predictions from GPT-4o. This improvement stems from the fine-tuned model's better alignment with class semantics and decision boundaries.

Furthermore, the accuracy drop due to compression remains minimal—within 0.6%—demonstrating the effectiveness of our method. On the Firefly dataset, despite the label prompt reaching 5.2k tokens and being compressed into a single token (achieving a 15× compression ratio), our method incurs only a 0.3% performance degradation while achieving a 6.9× inference speedup—an impressive result.

In the table-based QA task, evaluated on TableBench, which contains multi-table inputs, we report average metrics due to varying input complexity. GPT-4o achieves the highest accuracy, with our compressed version trailing by only 0.6%. This slight degradation is attributable to the limitations of Qwen2.5-7B in tabular reasoning and the inherent difficulty of compressing sparse table fields—a key challenge that renders existing compression baselines ineffective in this setting.

For tool-use scenarios, we evaluate on the xLAM dataset, which involves over 30k tools with descriptions totaling more than 3 million tokens. Such context lengths far exceed the model's input capacity, necessitating retrieval-based selection of the top-10 relevant tools, while ensuring the correct tool is included. Results show that non-compressed GPT-4o and LLaMA3.1-8B perform comparably. Our compressed method trails by 3.7% in accuracy. This gap is largely due to retrieval errors introduced during compression; assuming an optimistic 96% retrieval accuracy, the upper-bound performance would be 94.2%, which closely matches our compressed result. Notably, our method supports compression to as few as 13.3k tokens with strong performance. Latency measurements are based on VLLM-optimized benchmarking, providing a reliable view of relative speedups.

Compared with other compression baselines, our method achieves significantly superior performance—especially in large-tool scenarios. Hard compression methods such as LongLLMingua suffer from format degradation, leading to poor inference results despite high compression ratios. Among soft compression baselines, AutoCompressors, Gist, and 500xCompressor show moderate performance gains; however, AutoCompressors disrupt structural integrity due to recursive segmentation, and Gist/500xCompressor, relying solely on SFT-derived embeddings, incur information loss. In contrast, our approach fully reconstructs the original input during QA pair construction, ensuring minimal information loss. In the xLAM setting, where over 30k tools must be encoded, none of the baselines can incorporate all tool descriptions in a single input, resulting in severely degraded performance. Our method, by traversing and integrating all tool descriptions during QA construction, successfully preserves complete semantic content and maintains high performance.

## 5 DISCUSSION

### 5.1 HOW PROMPT LENGTH AND COMPLEXITY AFFECT THE COMPRESSION EFFECT

Our method is theoretically capable of handling prompts of arbitrary length, but does the compression effect vary with different prompt lengths? We observed this issue on the XLAM dataset. The XLAM dataset contains over 30,000 tool descriptions, with a total length exceeding 300,000 tokens. To study the impact of prompt length, the experimental setup was as follows:

1. A random tool category was selected to ensure the concatenated prompt length was under 8k. In this case, during the non-compressed test, all tool descriptions were concatenated before the query.

2. When the prompt length exceeded 8k, only the retrieved tool descriptions were concatenated in the non-compressed scenario.

From the results shown in Figure2a, we can observe three distinct segments:

- In the 0-0.5k range, compression and non-compression results were essentially the same.

- In the 1k-5k range, compression and non-compression results were also similar, but slightly worse than in the 0-0.5k range.

- For lengths greater than 5k, both compression and non-compression results declined as the length increased. This is understandable: longer prompts require the model to select the correct tools, which is a more demanding task. As the prompt length increases, the noise in the context also increases, and this noise reduces the performance.

For prompts longer than 8k, the compression effect was significantly better than non-compression. This is because non-compression depends on the accuracy of retrieval, as shown in Figure2a. When using ground truth, the performance can reach 98%, which demonstrates the critical impact of noise.

## 5.2 THE MORE THE BETTER?

Intuitively, the more tokens used for compressed representation, the larger the representational space and the better the performance. To verify this hypothesis, we experimented with using 1 to 20 tokens for representation across different tasks, and the final results are shown in Figure2b. We can observe that overall, as the number of tokens increases from 1 to 10, almost all tasks show an upward trend. However, further increasing the number of tokens does not lead to a significant improvement in performance. We can conclude that more tokens for compressed representation are not necessarily better; using just a few or even a single token is generally sufficient to meet the task requirements.

## 5.3 CAN UNSTRUCTURED PROMPT SCENARIOS BE USED

Although at the beginning of this paper we stated that our method focuses on structured prompts, theoretically, our method can handle prompts of any form. We selected three single-document-related tasks from Longbench to verify the effectiveness of our method when using documents as prompts. The experimental results are shown in Table 2. We compared two types of document tasks: retrieval-related methods and compression-related methods, following the experimental setup in Jiang et al. (2024). From Table 2, we can see that our method performs comparably to the current best prompt compression method (Jiang et al., 2024) and the original prompt. In the settings for Summa. and FewShot self-learning, our method even outperforms them, possibly because the compressed long documents have better denoising effects, enabling better learning of attention in the latent space with shorter contexts.

Table 2: Our method's performance in unstructured prompt tasks (selecting single-document relevant tasks in Longbench)

| Methods | SingleDoc | Summm. | FewShot |
|---|---|---|---|
| **Retrieval-based Methods** | | | |
| BM25 | 0.301 | 0.212 | 0.195 |
| SBERT | 0.338 | 0.259 | 0.235 |
| OpenAI | 0.343 | 0.247 | 0.324 |
| LongLLMLingua $\gamma_k$ | 0.378 | 0.269 | 0.663 |
| **Compression-based Methods** | | | |
| Selective-Context | 0.162 | 0.244 | 0.157 |
| LLMLingua | 0.224 | 0.245 | 0.612 |
| LongLLMLingua | 0.399 | 0.274 | 0.698 |
| Original Prompt | 0.397 | 0.265 | 0.670 |
| Zero-shot | 0.156 | 0.156 | 0.407 |
| LDPC(our's) | 0.393 | 0.270 | 0.668 |

## 5.4 IS PARALLEL CORPUS NEEDED, AND SHOULD ALL CONTENT BE COVERED

In the methods section, we detail how to use natural language to describe the prompts we aim to compress, primarily organizing them in a QA format. There are two issues we need to discuss in detail here: firstly, when constructing QA pairs for the same query, whether it is necessary to simultaneously construct parallel corpora for both compressed and non-compressed prompts; secondly, whether the QA pairs need to cover all the content in the prompt. To find the answers, we conducted experiments on Firefly and TableBench. For instance, to verify coverage, in the context of table

Table 3: Ablation experiments on parallel corpora and coverage.

| | Acc |
|---|---|
| Firefly | 0.887 |
| w/o all convered | 0.863 |
| w/o parrallel | 0.851 |
| TabelBench | 0.655 |
| w/o all convered | 0.644 |
| w/o parrallel | 0.626 |

QA, 'w/o all covered' indicates constructing QA pairs for each row of the table, and on this basis, 'w/o parallel' indicates not using parallel corpora. Table 3 shows the results, from which we can see that coverage positively impacts the results—the more comprehensive the coverage, the more sufficient the representation, and the better the performance. Parallel corpora are also crucial; if we remove the parallel corpora, the performance drops significantly because parallel corpora further align the space representation of the compressed tokens and the original prompts

## 6 RELATED WORK

**Based on Information Entropy** Empirical studies have shown that the performance of LLMs diminishes with less effective information in a prompt (Bai et al., 2024). Furthermore, the placement of relevant information within a prompt significantly influences performance (Wu et al., 2022). According to Liu et al. (2024a), LLMs struggle more with comprehending information located in the middle of a prompt than with information at the edges.

**Retrieval Based** Sparse Retrieval Methods Methods like BM25 determine the relevance between queries and documents based on n-gram information. Dense Retrieval Methods These methods assess relevance in latent space using embedding models (Reimers, 2019; Xiao et al., 2024; Günther et al., 2023) and reranker models (Xiao et al., 2024). Recently, Jiang et al. (2023b) introduced an unsupervised dense retrieval method that leverages traditional compression algorithms like gzip and k-nearest neighbors.

**Based on Compression** Hard prompt compression involves directly modifying natural language prompts to eliminate redundancy. Techniques include token pruning and merging, which require model fine-tuning or intermediate inference signals and have been primarily explored with BERT-scale models (Goyal et al., 2020; Kim & Cho, 2020; Modarressi et al., 2022; Bolya et al., 2022). Filtering-based approaches such as SelectiveContext estimate token importance using information entropy and remove less informative tokens (Li et al., 2023). Paraphrasing methods like Nano-Capsulator rephrase prompts to shorter yet semantically equivalent versions (Chuang et al., 2024). In contrast, soft prompt compression encodes prompts into continuous vectors. Representative soft prompt tuning methods, including GIST (Mu et al., 2024), AutoCompressor (Chevalier et al., 2023), and ICAE (Ge et al., 2023), require fine-tuning the LLM parameters, which suits domain-specific settings but is not directly applicable to black-box LLMs.

## 7 CONCLUSION

This paper proposes a prompt compression method based on language description to address the issue of high inference costs associated with structured prompts of arbitrary length. Compared to other compression methods, it achieves extreme compression while maintaining comparable performance, and the implementation is simple and straightforward.

## ETHICS STATEMENT

All data used in the experiments were obtained from publicly available sources or datasets with proper licenses. Our work does not involve any human subjects or animal experiments. We acknowledge the potential societal impacts of deploying language models, particularly in areas such as misinformation, bias, and fairness. We have made efforts to minimize these risks by carefully curating the datasets and ensuring that the methods are designed to avoid reinforcing harmful biases.

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
