# OpenReview forum: "StructZip: Compressing Large-Scale Structured Prompts to One Token via Learning Natural Language Descriptions"
_ICLR.cc/2026/Conference — Submitted to ICLR 2026_

### Official Review · Reviewer_sQBY · 2025-10-25

**Soundness:** 2
**Presentation:** 2
**Contribution:** 2
**Rating:** 2
**Confidence:** 3

**Summary:**

This paper introduces StructZip, a framework for large structured prompt compression (LSPC), aiming to compress long, rigidly formatted inputs (e.g., JSON schemas, API docs, tables) into a single special token that elicits equivalent semantic knowledge during inference.

**Strengths:**

- Tackles a genuinely underexplored but highly practical problem — compressing structured prompts rather than unstructured text, which previous works largely ignored.
- The idea of one-token elicitation for structured memory could have major implications for tool-using LLM agents and long-context efficiency.

**Weaknesses:**

- The framework lacks a formal information-theoretic model linking “QA-pair training” to the capacity of the compressed token.
For example, there is no bound on information retention or parametric storage entropy — how much structured data can truly be embedded into a token?
- GPT-4o is used as an untrainable reference, but StructZip models (Qwen2.5-7B, Llama3.1-8B) are fine-tuned — the comparison thus conflates compression with model specialization.
- Training involves converting millions of structured entries into QA pairs — the paper does not quantify the data explosion or compute cost of this “unzipping” process.
- Many implementation details (QA sampling ratios, SFT mixture percentage, optimizer configs) are omitted or buried in appendices.

**Questions:**

See above.

---

### Official Review · Reviewer_dqX1 · 2025-10-28

**Soundness:** 1
**Presentation:** 1
**Contribution:** 1
**Rating:** 0
**Confidence:** 4

**Summary:**

Defines the Large Structured Prompt Compression (LSPC) problem as using long, dense, and structure inputs (tables, JSON schemas, etc.) without paying inference token costs. Proposes StructZip which is comprised of two stages: (1) unzip the structure into reversible natural-language QA pairs via fixed templates, and (2) fine-tune the model on the QAs in a next token prediction fashion combined with a general SFT corpus while introducing a single new token whose learned embedding serves as a "key" to the compressed knowledge. During inference time, this token is used as a replacement for the original content.

**Strengths:**

- The paper covers an important and interesting open problem in LLM agents.

**Weaknesses:**

## Presentation Issues
- The paper uses `\citep{...}` instead of `\cite{...}` throughout.
- Line 183: my knowledge → our knowledge. Many parts in the paper require changes of this nature.
- Reduce informal phrasing. Example, Line 250: “Here are our experimental results across three different tasks” → We report results across three tasks.
- Improve table captions and column headers for precision and consistency (units, symbols, and ordering).

## Major Concerns
- I am highly skeptical of the results in Table 1, which reports unusually strong baselines for open models relative to GPT-4o (e.g., Chinese text classification ~90% vs ~72% for GPT-4o; English text classification where Llama-8B surpasses GPT-4o). This is plausible only under specific evaluation and prompting choices, which is not specified by the paper.
- The approach is a bit too close to recent context compression / learned representations, but many such works are not cited (e.g., [1] is notable). I do not see a big difference in the approach of StructZip to existing methods.

## Other Weaknesses
- Besides tool calling, text classification and some table QA setups are not compelling test beds for millions → one token compression. Given that the chief motivation is for AI agents and tool calling, the study should include harder settings where structured context is truly necessary and cannot be trivially summarized (e.g., large tool catalogs used by agents across tasks, schema evolution, compositional multi-table reasoning).

## Minor Comments
- The “w/” vs “w/o” rows mix models and settings and are hard to parse.

Overall, the paper is significantly below the acceptance standards of a typical ICLR paper and requires major revisions for reconsideration.


[1] Eyuboglu, S., Ehrlich, R., Arora, S., Guha, N., Zinsley, D., Liu, E., Tennien, W., Rudra, A., Zou, J., Mirhoseini, A., & Ré, C. (2025). *Cartridges: Lightweight and General-Purpose Long Context Representations via Self-Study.*

**Questions:**

- Why wouldn’t simple next-token prediction on the original structured content (or minimally re-formatted to plain text) work as well as the QA formulation? An ablation study would be necessary. Similarly, ablations on whether including general SFT data is necessary would be nice.
- The QA template set looks finite and hand-designed. Were templates chosen by trial-and-error to maximize results?

---

### Official Review · Reviewer_eFqv · 2025-10-30

**Soundness:** 3
**Presentation:** 2
**Contribution:** 2
**Rating:** 4
**Confidence:** 3

**Summary:**

This paper addresses the challenge of inference inefficiency when large language models (LLMs) process large-scale structured prompts, such as API documentation or complex tables. The authors introduce **StructZip**, a two-stage framework that first "unzips" a large structured prompt into a comprehensive set of semantically equivalent natural language question-answer (QA) pairs. Second, it fine-tunes an LLM on these QA pairs mixed with general-purpose instruction data, thereby encoding the structured information into the model's parameters. At inference, this embedded knowledge can be elicited by a single special token. Experiments across table question answering, tool-use, and text classification tasks demonstrate that StructZip can compress prompts of millions of tokens into one, achieving performance nearly on par with using the full, uncompressed prompts.

**Strengths:**

This paper has the following strengths:
1. Clear Problem Definition: The paper clearly identifies and motivates a significant, practical problem: the failure of existing compression techniques, designed for unstructured text, to handle information-dense, syntactically rigid structured data.
2. Comprehensive Task Evaluation: The authors validate **StructZip** across three distinct and representative tasks involving structured data: text classification (Sec. 4.1.1 ), table question-answering (Sec. 4.1.2 ), and tool invocation (Sec. 4.1.3 ), demonstrating the method's broad applicability.
3. Comprehensive discussion and ablations: The paper analyzes compression–performance trade-offs (Sec. 5), token count effects (Fig. 2b), and QA pair coverage impacts (Table 3).

**Weaknesses:**

This paper also has the following weaknesses:
1. Limited reproducibility details: Training hyperparameters, fine-tuning epochs, optimizer, and batch sizes are omitted (Sec. 3.2). No code or pseudo-code provided for QA generation (“fully reversible” process claimed without evidence).
2. Insufficient methodological clarity: The methodological description is overly concise, making it difficult to fully understand how structured data are transformed and compressed into the proposed token-based representation.
3. Limited technical contribution beyond dataset construction: Based on the description, the framework's primary contribution appears to be a specialized data curation pipeline that is then consumed by a SFT process. This raises a concern about the technical novelty of the work, as the method could be characterized as a clever application of SFT rather than a fundamentally new compression technique. (If this interpretation is incorrect, I would appreciate clarification on the novel technical components beyond the data generation.)
4. Lack of fidelity of "Unzipping": The method relies on the strong claim that the QA generation is "fully reversible" and "preserves the complete semantic and structural information" (Sec. 3.1 ). This is not empirically validated. It is unclear how fidelity is ensured for extremely complex, nested data. A qualitative analysis of potential information loss during this "unzipping" would be beneficial.

**Questions:**

Based on the identified weaknesses and aspects that are not entirely clear to me, I propose the following questions:
1. Regarding the "Structured Information Decoding" stage (Sec. 3.1), could you elaborate on the specific methodology used to generate the "diverse range of QA pairs"? For instance, were they created using rule-based templates or by prompting an auxiliary LLM, and what steps were taken to validate that this "unzipping" process is "fully reversible" and preserves the complete information from the original structured prompt?
2. The paper demonstrates strong results on static structured prompts. How does the StructZip framework handle dynamically changing data, such as an evolving API documentation? If a minor change is made to the source data, is it necessary to perform the entire SFT process again, or does the method support more efficient, incremental updates to the parametric memory?
3. Table 1 shows that StructZip without the compressed token (“w/o”) still achieves strong performance, sometimes close to the compressed variant. Could the authors clarify the reason behind this observation? Does this imply partial information retention in the base fine-tuned model or an inherent alignment effect from the SFT corpus?
4. The analysis on the number of compressed tokens (Sec. 5.2, Figure 2b) shows that performance generally saturates after ~10 tokens. However, for some tasks like TNEWS and Firefly, there appears to be a slight performance degradation when increasing the token count to 20. Do the authors have a hypothesis for this observation?

---

### Meta-Review · Area_Chair_3yQR · 2025-12-22

**Summary:**

The reviewers raise significant concerns regarding the paper's methodological novelty, empirical validity, and technical clarity. At its core, StructZip amounts to a QA-based data curation pipeline followed by standard supervised fine-tuning, rather than introducing a genuinely novel compression method. Empirically, the results are unconvincing: performance gains over strong baselines like GPT-4o appear implausible in the absence of a clearly defined and rigorous evaluation protocol, and the selected tasks fail to justify the necessity of extreme  compression ratios. Furthermore, the paper overlooks relevant prior work on learned context compression, weakening its positioning within the existing literature. On the presentation side, the manuscript suffers from informal language, inconsistent notation, and poorly designed tables. In light of these substantial shortcomings, I recommend rejection.

**Reviewer Concerns:**

Since the authors did not submit a rebuttal, none of the reviewers’ concerns were addressed. All issues raised remain outstanding, including the lack of methodological novelty, insufficient empirical validation, missing implementation details, implausible performance claims, inadequate justification for the use of extreme compression ratios, omission of relevant prior work, and weaknesses in presentation.

**Reviewer Scores:**

Since the authors did not submit a rebuttal, there was no new information or clarification provided that could have addressed the reviewers' concerns. Consequently, I believe none of the reviewers would have changed their scores.

---

### Decision · Program_Chairs · 2026-01-26

Reject